PERSPECTIVE

Applied and Environmental Science

# Introducing the Mangrove Microbiome Initiative: Identifying Microbial Research Priorities and Approaches To Better Understand, Protect, and Rehabilitate Mangrove Ecosystems

Sarah M. Allard,a Matthew T. Costa,b Ashley N. Bulseco,c Véronique Helfer,d Laetitia G. E. Wilkins,e Christiane Hassenrück,f,g Karsten Zengler,a,h,i Martin Zimmer,d,j,k Natalia Erazo,b Jorge L. Mazza Rodrigues,l Norman Duke,k,m Vânia M. M. Melo,n Inka Vanwonterghem,o Howard Junca,p Huxley M. Makonde,q Diego Javier Jiménez,r Tallita C. L. Tavares,s Marco Fusi,t,u Daniele Daffonchio,u Carlos M. Duarte,u Raquel S. Peixoto,e,v Alexandre S. Rosado,l,v Jack A. Gilbert,a,b,i Jeff Bowmanb,i

aDepartment of Pediatrics, School of Medicine, University of California San Diego, La Jolla, California, USA
bScripps Institution of Oceanography, University of California San Diego, La Jolla, California, USA
cMarine Science Discipline, Eckerd College, St. Petersburg, Florida, USA
dDepartment of Ecology, Leibniz Centre for Tropical Marine Research, Bremen, Germany
eGenome and Biomedical Sciences Facility, University of California Davis, Davis, California, USA
fTropical Marine Microbiology, Leibniz Centre for Tropical Marine Research, Bremen, Germany
gMicropaleontology-Paleoceanography, Center for Marine Environmental Sciences, University of Bremen, Bremen, Germany
hDepartment of Bioengineering, University of California San Diego, La Jolla, California, USA
iCenter for Microbiome Innovation, University of California San Diego, La Jolla, California, USA
jUniversity of Bremen, Faculty 02 Biology/Chemistry, Bremen, Germany
kIUCN SSC-Mangrove Specialist Group
lDepartment of Land, Air, and Water Resources, University of California Davis, Davis, California, USA
mCentre for Tropical Water and Aquatic Ecosystem Research, James Cook University, Townsville, Australia
nMicrobial Ecology and Biotechnology Laboratory, Federal University of Ceará, Fortaleza, Ceará, Brazil
oAustralian Centre for Ecogenomics, The University of Queensland, Brisbane, Australia
pRG Microbial Ecology: Metabolism, Genomics & Evolution, Microbiomas Foundation, Chia, Colombia
qDepartment of Pure & Applied Sciences, Technical University of Mombasa, Mombasa, Kenya
rMicrobiomes and Bioenergy Research Group, Department of Biological Sciences, Universidad de los Andes, Bogotá, Colombia
sMarine Sciences Institute, Federal University of Ceará, Fortaleza, Ceará, Brazil
tEdinburgh Napier University School of Applied Sciences, Edinburgh, United Kingdom
uRed Sea Research Center, BESE, Biological and Environmental Sciences and Engineering Division, KAUST, King Abdullah University of Science and Technology, Thuwal, Saudi Arabia
vFederal University of Rio de Janeiro (UFRJ), Rio de Janeiro, Brazil

Address correspondence to Sarah M. Allard, smallard@health.ucsd.edu, or Jeff Bowman, jsbowman@ucsd.edu.

As the global footprint of mangroves and their associated ecosystem services diminish, this Perspective introduces the Mangrove Microbiome Initiative and outlines 3 research priorities and 3 approaches to advance the field of mangrove microbiome research.

**ABSTRACT** Mangrove ecosystems provide important ecological benefits and ecosystem services, including carbon storage and coastline stabilization, but they also suffer great anthropogenic pressures. Microorganisms associated with mangrove sediments and the rhizosphere play key roles in this ecosystem and make essential contributions to its productivity and carbon budget. Understanding this nexus and moving from descriptive studies of microbial taxonomy to hypothesis-driven field and lab studies will facilitate a mechanistic understanding of mangrove ecosystem interaction webs and open opportunities for microorganism-mediated approaches to mangrove protection and rehabilitation. Such an effort calls for a multidisciplinary and collaborative approach, involving chemists, ecologists, evolutionary biologists, microbiologists, oceanographers, plant scientists, conservation biologists, and stakeholders, and it requires standardized methods to support reproducible experiments. Here, we outline the Mangrove Microbiome Initiative, which is focused around three urgent priorities and three approaches for advancing mangrove microbiome research.

**KEYWORDS** ecosystem rehabilitation, ecosystem services, mangrove, microbiome, rhizosphere

## INTRODUCTION: GLOBAL ROLE OF MANGROVES AND THEIR ASSOCIATED MICROBIOMES

Mangroves, intertidal forests along tropical and subtropical coasts, are hot spots of productivity and biodiversity. These ecosystems yield valuable services for humanity, including cultural and religious value (1), habitat for fisheries species (2), plant products including timber, filtration of terrestrial runoff, and coastline stabilization against storm impacts (3, 4). Globally, mangroves are significant carbon sinks (4), mitigating climate change by removing atmospheric greenhouse gases through sequestration of organic matter in above- and below-ground biomass. Ultimately, the mangrove ecosystem buries autochthonous and allochthonous detritus in anoxic, saline sediments, where this "coastal blue carbon" can remain stable for millennia (5, 6). Many of the ecological functions that underpin these services are carried out or supported by the microorganisms that comprise the mangrove microbiome, including bacteria, archaea, fungi, and protists.

Despite their economic and ecological importance, mangroves are threatened globally (7), especially by coastal development and pollution (8), and potentially by projected sea level rise (9, 10). Research to uncover the microbe-mangrove interactions that maintain ecosystem services and resilience under changing conditions is urgently needed for successful conservation and rehabilitation (10), making the nascent study of mangrove microbiome functions a high priority (8). As we enter what the United Nations has designated the Decade of Ocean Science for Sustainable Development as well as the Decade on Ecosystem Restoration, building international collaboration working toward science-based management of coastal ecosystems is an extremely timely endeavor (11, 12).

This Perspective proposes microbiological research objectives and approaches to meet the mangrove management challenges of the 21st century. We have formed the Mangrove Microbiome Initiative (MMI), an international network of researchers advancing mangrove microbiome research through collaboration, discussion, and advocacy. The aim of this platform is to facilitate collaborative work and knowledge sharing among all researchers who wish to participate, strengthening our collective efforts toward understanding, protecting, and rehabilitating these important ecosystems. Research has so far only scratched the surface of understanding the diversity, function, and connectivity of mangrove microbiomes. Recent developments in -omics techniques and bioinformatic pipelines have changed the way we look at genes, species, and communities, opening new windows into mangrove ecology. A more complete understanding of mangrove-microbe interactions will support efforts to rehabilitate mangrove forests and sustain ecosystem services in the face of increasing anthropogenic stress. Here, we identify three priority research areas for mangrove microbiome research (priority 1 [P1], P2, and P3) and discuss three approaches to advancing the field (approach 1 [A1], A2, and A3) (Fig. 1).

## PRIORITY RESEARCH AREAS

**P1. Characterizing mangrove microbiomes across scales in a changing world.** Understanding and predicting the influence of global change on the mangrove microbiome is an important goal and a great challenge that offers opportunities to protect, manage, and mitigate impacts to threatened mangroves. At present, much of the work characterizing microbial communities in mangroves has been descriptive and limited in temporal and spatial range. While descriptive studies provide an important foundational understanding of the mangrove microbiome, there is a need to advance the field toward hypothesis-driven observational and experimental research to establish the mechanisms that underlie mangrove-microbe symbiosis in these variable and far-flung ecosystems. Achieving a mechanistic understanding requires detailed quantification of biotic (e.g., plant taxonomy, anatomy, and sediment fauna) and abiotic (e.g., temper-

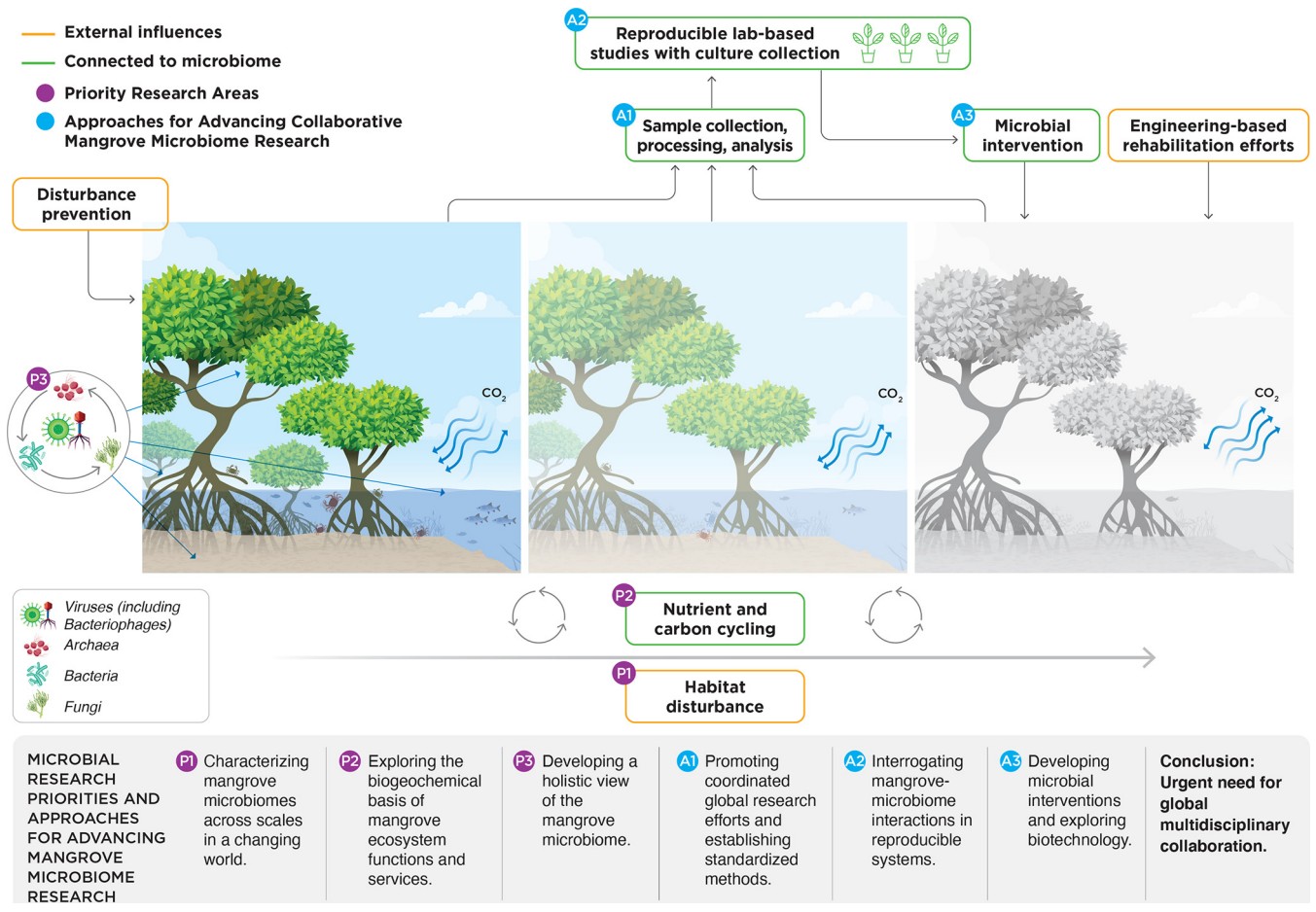

**FIG 1** Microbial research priorities and approaches to better understand, protect, and rehabilitate mangrove ecosystems.

ature, salinity, tidal amplitude and frequency, and level of pollution) variables that can influence the composition and functions of the microbiome. Dynamic spatiotemporal factors such as fluctuating air exposure times, oxygen concentrations (13), and salinity levels (14), in addition to seasonal variations in rainfall (15), can affect the microbiome, thereby influencing mangrove productivity. At a fine spatial scale, vicinity to vegetation and crab burrows can also affect microbial metabolism (16, 17). To better understand these processes, sophisticated experimental designs, new technologies and analytical approaches, and directed intervention studies are required, as discussed below in Approaches.

**P2. Exploring the biogeochemical basis of mangrove ecosystem functions and services.** Mangroves support diverse communities of microorganisms in sediment layers, in the water column, and in and on their tissues (e.g., biofilms on mangrove roots) (18), and these communities play crucial roles in mangrove biogeochemistry and nutrient cycling (19, 20). Indeed, a large fraction of the carbon turnover in these ecosystems is carried out by sediment microbial heterotrophs (21). Bacterial oxygen consumption and sulfate reduction generate chemical conditions in mangrove sediments that slow organic matter turnover, favoring the establishment of a net carbon sink (22). In addition, microbial metabolism along sediment redox gradients drives the production and consumption of methane and nitrous oxide (19, 23), potentially resulting in net sources of these greenhouse gases (24). Microbes play a critical role in nitrogen cycling in mangrove sediments through a broad array of processes, including fixation, denitrification, and anammox (anaerobic ammonium oxidation) (19, 25). They contribute to remineralizing, and solubilizing otherwise unavailable phosphorus (26),

thereby mediating the availability and fluxes of nutrients that can potentially limit mangrove plant productivity (27, 28). This productivity fuels plant-product-based eco-system services and provides the basis of the mangrove ecosystem food web (29–31), which feeds the valuable services of fisheries production and mangrove ecotourism (2, 32). In addition, microbes on root and leaf surfaces make micronutrients available, can provide defense against pathogens, and launch decay processes upon senescence (18, 33). While the relevance of these microbial processes to biogeochemical cycling and to the associated services of carbon sequestration and nutrient regulation have been demonstrated, mechanistic and predictive understanding is still in its infancy. The Approaches section below discusses how standardized, experimental, and process-based studies will move this field toward predictive understanding.

**P3. Developing a holistic view of the mangrove microbiome.** Although there are numerous studies on mangrove bacterial communities (e.g., 18, 34, 35), there are relatively few studies on fungi, protists, archaea, and viruses (including phage and eukaryotic viruses). This knowledge gap is further complicated by the complex struc-ture of the mangrove root system. Many metazoan inhabitants of this system, including sponges, oysters, clams, and cockles, have their own distinct microbiomes that also contribute to ecosystem functioning. Future work must take into account the full extent of the taxonomic, functional, and structural diversity of the mangrove forest. It is essential, for example, to explore the evolution, ecology, and physiology of mangrove-associated microbial eukaryotes. Fungi and protists in particular are thought to play a significant role in the ecology of mangrove forests (36) and can be bioindicators of pollutants (37, 38). Unlike prokaryotes, their activity and function in natural ecosystems are not based on a large flexibility of their metabolic capacities but on the exploration of innovations in their structural complexity and behaviors (39). Fungi are capable of converting complex organic compounds into more easily accessible forms and provide infrastructure (i.e., fungal highways) along which bacteria migrate to areas with pref-erential nutrients, substrates, and redox conditions (40, 41). Among protists, thraus-tochytrids, known for their saprobic capabilities (42) and their ability to degrade highly recalcitrant organic matter, also play an essential role in organic matter turnover in mangrove ecosystems and can be an important food source for detritivores (43). Archaea represent another underexplored group in mangrove ecosystems. Ammonia-oxidizing archaea (and their bacterial counterparts) are common in mangrove sedi-ments, as are methanogenic archaea. Beyond simple observations of community structure (44), however, very little is known about the function of mangrove-associated archaeal communities. Approaches to future mangrove microbiome research should be more inclusive of nonbacterial microorganisms to facilitate a more holistic understand-ing of the ecosystem.

## APPROACHES FOR ADVANCING COLLABORATIVE MANGROVE MICROBIOME RESEARCH

**A1. Promoting coordinated global research efforts and establishing standard-ized methods.** Coordinated global efforts to ensure data quality and comparability are essential (45), as this enables integration for meta-analysis as well as more distributed analytical capability. The MMI will promote coordinated international mangrove mon-itoring networks and contribute to standardization of sampling, analytical procedures, and data archiving.

Robust experimental design (e.g., sufficient and appropriate replication) and sam-pling protocols are crucial to move beyond anecdotal observation and reduce the masking effect of confounding factors. Representative sampling requires consideration of the appropriate scale and target community (e.g., epibionts and endobionts) and how to address variation within sampling units (e.g., combining subsamples). Special attention should be paid to factors influencing microbial habitat, such as sediment depth (34, 46), light exposure (47), variety of root structures among and within mangrove tree species (48), and leaf senescence.

To embed our efforts in a larger scope beyond mangrove ecosystem boundaries, standards from existing initiatives (e.g., the Earth Microbiome Project) should be adapted for the generation of mangrove microbiome data (49). Our current toolbox to study the mangrove microbiome includes modern -omics techniques (metagenomics, metatranscriptomics, metaproteomics, and metabolomics), physiology and biochemistry (cultivation, colonization, and metabolic modeling), imaging (three-dimensional [3D] tomography, histology, electron microscopy, superresolution microscopy, and mass spectrometry), hypothesis-driven field studies, and the use of reproducible laboratory systems. The MMI platform will be used to circulate standard protocols adjusted to mangrove research for quality assurance and reproducibility. We envision a sustainable collaborative research approach where samples are collected and stored for a broad scope of future applications and where standardized metadata (including contextual information, analytical protocols, and bioinformatic pipelines) are accessible. Some existing international initiatives can be leveraged to meet this goal. For example, infrastructure to meet these aims can be supported by the establishment of mangrove monitoring programs in ILTER (International Long-Term Ecosystem Research) sites and the expansion of the ILTER network to cover sites from a broad range of mangrove environmental settings around the world (50). We recommend strict adherence to existing checklists for data archiving (51) and additional submission of nonmandatory parameters to improve compliance with the FAIR principle (Findable, Accessible, Interoperable, and Reusable [52]). The MMI will develop and promote the use of essential variables in coordination with GOOS (Global Ocean Observing System [53, 54]) to support concerted documentation and monitoring of the mangrove microbiome across spatial and temporal scales.

**A2. Interrogating mangrove-microbiome interactions in reproducible systems.** Mangrove ecosystems are threatened by a multitude of stressors, such as pollution, sea level rise, coastal development, and sediment salinization. There is an urgent need to better understand the impacts of these stressors on mangroves and their microbiomes. In contrast to observational studies, controlled lab-based experiments enable manipulation of specific disturbances and quantification of effects on microbial populations. Using new microbial ecology approaches and technologies, it is possible to predict and test response to perturbation and provide insight into the mechanisms behind these responses (55, 56). Controlled laboratory settings can yield reproducible results while eliminating environmental fluctuations and high costs often associated with field studies requiring large sample sizes (57). Model ecosystems with the potential to enable reproducible mangrove microbiome research range from highly controllable enclosed systems like EcoFABs (fabricated microbial ecosystems) (58, 59) to larger scale systems that introduce more variation and complexity (46, 60).

Generating robust synthetic microbial communities for use in these reproducible systems requires isolation of representative microorganisms, a historically challenging task. Successful approaches to reducing isolation bias include dilution to extinction (61, 62), encapsulation or separation (63, 64), and growth on chips (65, 66), all approaches that imitate the natural environment to a certain degree, e.g., by operating with low nutrient concentrations. Other top-down strategies, like dilution to stimulation (67) or targeting specific microbes based on gene content (68, 69), could be useful to develop specific mangrove-derived microbial consortia with desired functional roles (33), while automated cultivation procedures can lower cost and increase throughput. Expanding microbial culture collections from mangrove environments will be necessary to unravel beneficial plant-microbe interactions, including plant-health-promoting bacteria (33), and to provide necessary cultures for improving mangrove health in the future.

**A3. Developing microbial interventions and exploring biotechnology.** Specific threats to mangrove forests may be remediated or mitigated by manipulation of microorganisms. For example, *in situ* characterization of microbial biodegradation potential allows for the development of strategies for microbiome manipulation as a tool to prevent and/or mitigate oil impacts on mangroves (34, 37, 46, 70, 71), which are vulnerable to chronic oil spills (72). Of particular interest are oil-degrading, health-

promoting (ODHP) microbial consortia (33), which have dual functions: promoting oil degradation and improving ecosystem and plant health. Indeed, microbial consortia in mangrove sediments have been found to efficiently degrade oil, rendering them a potential resource of effective hydrocarbon-degrading bacteria that can be used as an inoculum for the purpose of bioremediation (73).

It is important to recognize that mangrove-health-promoting bacteria may have additional applications to agriculture and other systems. A recent study of the microbiome associated with propagules of the mangrove plant *Avicennia marina* in the Red Sea revealed plant-growth-promoting bacteria that enhance root development and mangrove establishment (42). In addition, bacterial strains isolated from mangroves have shown promise for salinity adaptation in agriculture (42) and for removal of cadmium and zinc from hazardous industrial residue (74), highlighting the biotechnological potential of mangrove-associated microbes to mitigate environmental impacts. By marshaling the full suite of modern -omics tools, there is great promise for the development of evidence-based ecosystem rehabilitation techniques for mangrove and agricultural ecosystem functioning.

## CONCLUSION: URGENT NEED FOR GLOBAL MULTIDISCIPLINARY ACTION

To advance the field of mangrove microbiome research and to facilitate protection and rehabilitation of these crucial ecosystems, there is an urgent need for global multidisciplinary collaboration that leads to action. Here, we have identified three research priorities and three approaches to advance the field, and we have committed to building a broad, collaborative network of researchers across disciplines, including chemists, ecologists, evolutionary biologists, microbiologists, oceanographers, plant scientists, conservation biologists, and government representatives. Global collaboration to establish universal protocols with a constantly expanding and versatile toolbox will facilitate the collection of valuable data simultaneously and across the globe. Testing hypotheses to elucidate microbial metabolisms that support mangrove rehabilitation is critically dependent on field experiments extending over the multiyear time scale of intervention success or failure and on the consistent measurement of defined variables important for mangrove health assessment. While such an investment may seem unattractive in a fast-moving field, documentation of long-term results will be a rare and valuable contribution to global mangrove restoration and rehabilitation efforts and will be beneficial for successfully designing mangrove ecosystems for the provisioning of particular ecosystem services. Furthermore, these approaches will be valuable, not only for mangrove ecosystems, but will also have the potential for application to other coastal ecosystems and even terrestrial agricultural systems in the future.

With the establishment of the Mangrove Microbiome Initiative (http://bmmo .microbe.net/mangrove-microbiome-initiative-mmi/) as part of the Beneficial Microbes for Marine Organisms (BMMO) network, we seek to bridge the breadth of knowledge from researchers focusing on the ecology and physiology of mangrove systems and those with expertise in microbiology, high-throughput molecular methods, and bioinformatics. We welcome any interested researchers working on the mangrove microbiome to join our network through our website. This network provides a platform to establish common goals and foster collaboration among groups working around the globe and to share not only technical expertise but also crucial advice for overcoming logistical barriers and enabling long-term research and rehabilitation success. For increased awareness and longevity of research and interventions, engagement with local communities and buy-in from decision-makers is essential. Furthermore, with the high cost associated with accessing remote locations, the challenging logistics of obtaining permits for research sites, and the importance of prioritizing just practices for the extraction of samples and data from field sites, collaborative approaches from multiple research groups in multiple regions and countries would be most efficient.

As the global footprint of mangroves and their associated ecosystem services continue to diminish, advancement of the field of mangrove microbiome research is urgently needed. Microorganisms have seldom been included in ecosystem manage-

ment plans and policy, but as our understanding of their importance in maintaining ecosystem health and enhancing resilience in the face of global change grows (75), it is crucial to acknowledge their role and the opportunities that they provide.

## ACKNOWLEDGMENTS

We thank Jonathan Eisen (ORCID https://orcid.org/0000-0002-0159-2197) and the microBEnet platform for the full and kind support provided to the BMMO network and its initiatives.

M.F. acknowledges funding from the Natural Environment Research Council (NERC grant NE/S006990/1).

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
