## [Reviewer comments · mSystems]

Introducing the Mangrove Microbiome Initiative: Identifying microbial research priorities and approaches to better understand, protect, and rehabilitate mangrove ecosystems

Sarah Allard, Matthew Costa, Ashley Bulseco, Véronique Helfer, Laetitia Wilkins, Christiane Hassenrück, Karsten Zengler, Martin Zimmer, Natalia Erazo, Jorge Rodrigues, Norman Duke, Vânia Melo, Inka Vanwonderghem, Howard Junca, Huxley Makonde, Diego Jimenez, Tallita Tavares, Marco Fusi, Daniele Daffonchio, Carlos Duarte, Raquel Peixoto, Alexandre Rosado, Jack Gilbert, and Jeff Bowman

Corresponding Author(s): Sarah Allard, University of California San Diego

Review Timeline:

Submission Date:	July 13, 2020
Editorial Decision:	August 13, 2020
Revision Received:	September 23, 2020
Accepted:	September 30, 2020

Editor: Jean-Baptiste Raina

Reviewer(s): Disclosure of reviewer identity is with reference to reviewer comments included in decision letter(s). The following individuals involved in review of your submission have agreed to reveal their identity: Janet K. Jansson (Reviewer #1); Stacey Trevathan-Tackett (Reviewer #2)

Transaction Report:

DOI: <https://doi.org/10.1128/mSystems.00658-20>

August 13, 2020

Dr. Sarah M. Allard
University of California San Diego
Department of Pediatrics
9500 Gilman Drive #0202
La Jolla, CA 92093

Re: mSystems00658-20 (Introducing the Mangrove Microbiome Initiative: Setting microbial research priorities for mangrove preservation and rehabilitation)

Dear Dr. Sarah M. Allard:

Both reviewers have found your Perspective timely, important and well written. They have suggested a few modifications, which should be considered before this manuscript is accepted.

Below you will find the comments of the reviewers.

To submit your modified manuscript, log onto the eJP submission site at <https://msystems.msubmit.net/cgi-bin/main.plex>. If you cannot remember your password, click the "Can't remember your password?" link and follow the instructions on the screen. Go to Author Tasks and click the appropriate manuscript title to begin the resubmission process. The information that you entered when you first submitted the paper will be displayed. Please update the information as necessary. Provide (1) point-by-point responses to the issues raised by the reviewers as file type "Response to Reviewers," not in your cover letter, and (2) a PDF file that indicates the changes from the original submission (by highlighting or underlining the changes) as file type "Marked Up Manuscript - For Review Only."

Due to the SARS-CoV-2 pandemic, our typical 60 day deadline for revisions will not be applied. I hope that you will be able to submit a revised manuscript soon, but want to reassure you that the journal will be flexible in terms of timing, particularly if experimental revisions are needed. When you are ready to resubmit, please know that our staff and Editors are working remotely and handling submissions without delay. If you do not wish to modify the manuscript and prefer to submit it to another journal, please notify me of your decision immediately so that the manuscript may be formally withdrawn from consideration by mSystems.

To avoid unnecessary delay in publication should your modified manuscript be accepted, it is important that all elements you upload meet the technical requirements for production. I strongly recommend that you check your digital images using the Rapid Inspector tool at <http://rapidinspector.cadmus.com/RapidInspector/zmw/>.

Corresponding authors may join or renew ASM membership to obtain discounts on publication fees.

Need to upgrade your membership level? Please contact Customer Service at Service@asmusa.org.

Sincerely,

Jean-Baptiste Raina

Editor, mSystems

Journals Department
Reviewer comments:

Reviewer #1 (Comments for the Author):

I enjoyed reading this perspective manuscript and agree that the topic is timely and important. The manuscript was very well written, and I only have a few minor comments and suggestions that I list below:

1. Figure: Some of the text is small and may be hard to read on the figure
2. More clarity should be provided about the MMI. Is it funded? Is the MMI an organization? an adhoc group? a call for funding? a consortium? I looked up the website, but it did not provide much information as it was 'under construction'.
3. Line 66: Are there no LTER sites currently for mangrove ecosystems?
4. Note: I recall that mangroves were intensively studied during the Deepwater Horizon spill. Do you have those refs if relevant?
5. Line 88 - provide some refs for examples for bacterial communities.
6. Line 97: ref for 'fungal highways'
7. line 105: apparently something is also known about archaea, not just bacteria based on previous sentences.
8. line 105 and elsewhere: The manuscript is rather dismissive of what is known from studies of bacteria. Maybe provide some examples of what is known and a figure?

Reviewer #2 (Comments for the Author):

A very interesting initiative and a nice first output for the network and will advance the mangrove microbiome field as a whole. The figure makes clear the connections across the priorities, but the text doesn't quite reflect this yet, possibly due to the sections being written by different people. Most of my comments are around making these priorities linked to or complementary of each other and to the title. For example, since the aim of the paper is to set the priorities, the research gaps need to be make clear - this is done in some sections well but not others. Making the format of each priority section similar in terms of how the gaps and the next-step recommendations are presented will also improve readability, flow and the links among the priorities.

The terms preservation and rehabilitation are in the title, yet the link in each priority to rehabilitation/restoration isn't always clear, and sometimes a stretch. For example, some of the priorities would generally advance the mangrove microbiome field as a whole rather than have specific connection to restoration. Resolving microbiome function re mangrove (or ecosystem) health doesn't have to be in the context of restoration either. Perhaps the title/paper can be broadened to encompass more than the restoration/rehab theme. This will cast a wider readership as well.

Specific comments:

L21: Please clarify a bit more the two pathways for sequestration: Carbon in biomass (is a pathway unto itself) and capture of carbon in soils (from external sources and the mangrove plants themselves via decay).

Priority A: Are the authors suggesting that observational studies are no longer important or valuable to understanding the mangrove microbiome? Or do observational studies still have a place in mangrove microbiome research? At the moment mangrove microbiome data are still nascent compared to other ecosystems like coral and sponges, and one could argue that the observational type studies still hold value for building the foundation for understanding the microbiome of a certain (underrepresented) condition/habitat/ecosystem/region in order to understand novel biodiversity and help build the hypothesis-based questions priority A is encouraging.

Priority B:

L71 - reference 16 seems to mostly focus on diazotrophs (and N-cycling) and doesn't speak to the diversity of microorganisms in mangrove ecosystems. Perhaps the Holguin et al 2001 review cited late or a paper such as the one below, though limited in its compartments and geographic scope, would speak to the biodiversity and biogeochem functions. Lin, X., Hetharua, B., Lin, L., Xu, H., Zheng, T., He, Z., & Tian, Y. (2019). Mangrove sediment microbiome: adaptive microbial assemblages and their routed biogeochemical processes in Yunxiao mangrove national nature reserve, China. *Microbial ecology*, 78(1), 57-69.

Can the authors provide more details on the current gaps for this section, similar to the gap / explicit research need presented in priority A?

The title for B doesn't capture all the content presented. For example, the link to ecosystem services isn't clear, other than a mention of carbon sink and GHG mitigation. What about nutrient cycling, pollution reduction and food-web support? The connection of this biogeochem paragraph to how it can help restoration and management efforts is also vague and not contextualised. Along with the previous comment, this priority could be reframed to focus less detail on the knowns (L72-

82), rather given the knowns, what are the gaps in what we know about biogeochemistry to promote X and how the microbiome can help answer that.

Priority C:

L95-98: please provide a reference for this phenomenon

There is some overlap with biogeochemistry themes in priorities B and C. The examples on the lack of knowledge of the eukaryotic members of the microbiomes would also fit in with the biogeochemical gaps. Some of those details can be moved to B with referencing to the other priorities, and perhaps this section could focus not only on the unknowns (and some knowns) of how the non-bacterial members of the microbiome function in the ecosystem or with the host, but also how these microbiome members interact with each other, eg micro-food web, competition/colonisation, etc. If there is nothing on this in the literature, perhaps this is where observational studies could be useful as a starting point.

Priorities B, C: Can the authors include suggestions as to how these priorities could be achieved as done in A? Providing examples across all priorities is good for consistency but could improve what the readership gleans from the paper and thereby cites it.

Priority D: It's only clear at the end of the paragraph that MMI was created to provide that platform for standardisation. A statement at the beginning of the second paragraph would be useful (L120) followed by how it will be done / what it will encompass. (can also reference to the conclusion or vice versa)

Priority E could be linked to B re restoration and management. Also the threat to mangrove health isn't clear - the introduction briefly mentioned pollution, but not clear what the threats to mangrove health are and how knowledge of microbiomes are critical to that understanding.

Priority F: Following on the above comments, the health connections looks like it's covered more here, so again integration across the priorities would be good to have. Here I see connections to B that can be made re restoration - or perhaps the case for restoration is more fitting in F. Since restoration and rehabilitation is an ongoing theme, perhaps it can be its own priority if the authors decide to broaden the title and scope of the paper.

L173-175: can a reference be added here or is this unpublished?

Conclusions: It seems like MMI has an established team which are mostly the co-authors - is this the mentioned collaborative network and closed to people joining, or is there a call for more collaborators or data using the framework outlined in D to build research capacity, questions and datasets?

Figure 1: Suggest having the description of E in one section to improve interpretation

We thank Reviewers 1 and 2 for their helpful feedback. In response to the below reviewer suggestions (black text), we have modified the text and figure as described below (blue text). In the marked up version of the manuscript, changes are underlined. In response to the reviewer suggestions for restructuring the manuscript, we have re-organized the text and figure into research priorities and approaches to advance the field. We believe that the manuscript is much improved.

Reviewer #1 (Comments for the Author):

I enjoyed reading this perspective manuscript and agree that the topic is timely and important. The manuscript was very well written, and I only have a few minor comments and suggestions that I list below:

Thank you very much for your kind comments and constructive feedback. Below are our responses to your suggestions.

1. Figure: Some of the text is small and may be hard to read on the figure

We have reorganized the text in the figure to be more easy-to-read. We have also updated the figure to reflect the new organization of the priorities and approaches.

2. More clarity should be provided about the MMI. Is it funded? Is the MMI an organization? an adhoc group? a call for funding? a consortium? I looked up the website, but it did not provide much information as it was 'under construction'.

We have clarified the text as follows (Lines 39-44), and we have also recently updated the website with more information about the goals of the group and instructions for how to join.

“We have formed the Mangrove Microbiome Initiative (MMI), an international network of researchers advancing mangrove microbiome research through collaboration, discussion, and advocacy. The aim of this platform is to facilitate collaborative work and knowledge-sharing amongst all researchers who wish to participate, strengthening our collective efforts towards understanding, protecting, and rehabilitating these important ecosystems.”

3. Line 66: Are there no LTER sites currently for mangrove ecosystems?

There are a limited number of LTER sites containing mangrove ecosystems. We have updated the text to discuss the need to increase the number and diversity of locations for additional sites. This can be seen in Lines 152-155.

4. Note: I recall that mangroves were intensively studied during the Deepwater Horizon spill. Do you have those refs if relevant?

The area impacted by the Deepwater Horizon oil spill has very limited mangrove cover, so the impact assessment focussed largely on saltmarshes (e.g., Silliman et al. 2012). However, there were some impacts on mangroves, so we have added a citation in section A3, line 193.

71. Lamendella R, Strutt S, Borglin S, Chakraborty R, Tas N, Mason OU, Hultman J, Prestat E, Hazen TC, Jansson JK. 2014. Assessment of the Deepwater Horizon oil spill impact on Gulf coast microbial communities. *Front Microbiol* 5.

5. Line 88 - provide some refs for examples for bacterial communities.

We have added 3 references as examples, and an ellipses to indicate that this is a subset of what is available.

6. Line 97: ref for 'fungal highways'

We have added 2 references here.

40. Kohlmeier S, Smits THM, Ford RM, Keel C, Harms H, Wick LY. 2005. Taking the Fungal Highway: Mobilization of Pollutant-Degrading Bacteria by Fungi. *Environ Sci Technol* 39:4640–4646.

41. Warmink JA, Nazir R, Corten B, van Elsas JD. 2011. Hitchhikers on the fungal highway: The helper effect for bacterial migration via fungal hyphae. *Soil Biol Biochem* 43:760–765.

7. line 105: apparently something is also known about archaea, not just bacteria based on previous sentences.

We have added some additional sentences to reflect this, in lines 116-120:

“Archaea represent another under-explored group in mangrove ecosystems. Ammonia oxidizing archaea (and their bacterial counterparts) are common in mangrove sediments, as are methanogenic archaea. Beyond simple observations of community structure (44), however, very little is known about the function of mangrove-associated archaeal communities.”

8. line 105 and elsewhere: The manuscript is rather dismissive of what is known from studies of bacteria. Maybe provide some examples of what is known and a figure?

We do believe that earlier studies of bacteria in mangroves using pre -omics techniques were valuable even if they were not able to do what we can today. We've added some language (in the intro, in sections P1, P2, and A1) that makes that point more diplomatically, that we have some decades of bacterial and fungal research in mangroves which made important observations and suggested some general fluxes and rates in a general way, but that now we have the tools to interrogate these systems with taxonomic breadth and precision and to move much closer to mechanistic understanding.

Reviewer #2 (Comments for the Author):

A very interesting initiative and a nice first output for the network and will advance the mangrove microbiome field as a whole. The figure makes clear the connections across the priorities, but the text doesn't quite reflect this yet, possibly due to the sections being written by different people. Most of my comments are around making these priorities linked to or complementary of each other and to the title. For example, since the aim of the paper is to set the priorities, the research gaps need to be make clear - this is done in some sections well but not others. Making the format of each priority section similar in terms of how the gaps and the next-step recommendations are presented will also improve readability, flow and the links among the priorities.

Thank you for your helpful feedback. We have reorganized the manuscript into research priorities and approaches, and we have added text to link the sections together more clearly. A team of authors went through the document together to ensure cohesiveness. We think the flow is much improved.

The terms preservation and rehabilitation are in the title, yet the link in each priority to rehabilitation/restoration isn't always clear, and sometimes a stretch. For example, some of the priorities would generally advance the mangrove microbiome field as a whole rather than have specific connection to restoration. Resolving microbiome function re mangrove (or ecosystem) health doesn't have to be in the context of restoration either. Perhaps the title/paper can be broadened to encompass more than the restoration/rehab theme. This will cast a wider readership as well.

We agree with this assessment and have changed the title to reflect a more broad range of goals. The reorganization better highlights these goals as well.

Specific comments:

L21: Please clarify a bit more the two pathways for sequestration: Carbon in biomass (is a pathway unto itself) and capture of carbon in soils (from external sources and the mangrove plants themselves via decay).

We have clarified the text, which now reads:

“Globally, mangroves are significant carbon sinks (4), mitigating climate change by removing atmospheric greenhouse gases through sequestration of organic matter in above- and below-ground biomass. Ultimately the mangrove ecosystem buries autochthonous and allochthonous detritus in anoxic, saline sediments, where this “coastal blue carbon” can remain stable for millennia (5, 6).”

Priority A: Are the authors suggesting that observational studies are no longer important or valuable to understanding the mangrove microbiome? Or do observational studies still have a place in mangrove microbiome research? At the moment mangrove microbiome data are still nascent compared to other ecosystems like coral and sponges, and one could argue that the observational type studies still hold value for building the foundation for understanding the microbiome of a certain (underrepresented) condition/habitat/ecosystem/region in order to understand novel biodiversity and help build the hypothesis-based questions priority A is encouraging.

We agree that observational studies are still valuable in mangrove microbiome research, and we have updated the text to reflect their importance (lines 58-62).

“While descriptive studies provide an important foundational understanding of the mangrove microbiome, there is a need to advance the field towards hypothesis-driven observational and experimental research to establish the mechanisms that underlie mangrove-microbe symbiosis in these variable and far-flung ecosystems.”

Priority B:

L71 - reference 16 seems to mostly focus on diazotrophs (and N-cycling) and doesn't speak to the diversity of microorganisms in mangrove ecosystems. Perhaps the Holguin et al 2001 review cited later or a paper such as the one below, though limited in its compartments and geographic scope, would speak to the biodiversity and biogeochem functions. Lin, X., Hetharua, B., Lin, L., Xu, H., Zheng, T., He, Z., & Tian, Y. (2019). Mangrove sediment microbiome: adaptive microbial assemblages and their routed biogeochemical processes in Yunxiao mangrove national nature reserve, China. *Microbial ecology*, 78(1), 57-69.

We have removed this reference and replaced it with the Holguin et al review, which we agree is a better fit here, and we have added more detail and additional references to this section (lines 81-90).

Can the authors provide more details on the current gaps for this section, similar to the gap / explicit research need presented in priority A?

This comment is addressed by the restructuring into research priorities and approaches.

The title for B doesn't capture all the content presented. For example, the link to ecosystem services isn't clear, other than a mention of carbon sink and GHG mitigation. What about nutrient cycling, pollution reduction and food-web support? The connection of this biogeochem paragraph to how it can help restoration and management efforts is also vague and not contextualised. Along with the previous comment, this priority could be reframed to focus less detail on the knowns (L72-82), rather given the knowns, what are the gaps in what we know about biogeochemistry to promote X and how the microbiome can help answer that.

We prefer to keep the titles concise but have changed “ecosystem services” to “ecosystem functioning and services” to better capture the content of the section. The change in title has expanded the scope of the manuscript beyond restoration and management, and the reorganization addresses these concerns as well.

Priority C:

L95-98: please provide a reference for this phenomenon

We have added 2 references here:

40. Kohlmeier S, Smits THM, Ford RM, Keel C, Harms H, Wick LY. 2005. Taking the Fungal Highway: Mobilization of Pollutant-Degrading Bacteria by Fungi. *Environ Sci Technol* 39:4640–4646.
41. Warmink JA, Nazir R, Corten B, van Elsas JD. 2011. Hitchhikers on the fungal highway: The helper effect for bacterial migration via fungal hyphae. *Soil Biol Biochem* 43:760–765.

There is some overlap with biogeochemistry themes in priorities B and C. The examples on the lack of knowledge of the eukaryotic members of the microbiomes would also fit in with the biogeochemical gaps. Some of those details can be moved to B with referencing to the other priorities, and perhaps this section could focus not only on the unknowns (and some knowns) of how the non-bacterial members of the microbiome function in the ecosystem or with the host, but also how these microbiome members interact with each other, eg micro-food web, competition/colonisation, etc. If there is nothing on this in the literature, perhaps this is where observational studies could be useful as a starting point.

This comment is addressed by the restructuring into research priorities and approaches.

Priorities B, C: Can the authors include suggestions as to how these priorities could be achieved as done in A? Providing examples across all priorities is good for consistency but could improve what the readership gleans from the paper and thereby cites it.

This comment is addressed by the restructuring into research priorities and approaches.

Priority D: It's only clear at the end of the paragraph that MMI was created to provide that platform for standardisation. A statement at the beginning of the second paragraph would be useful (L120) followed by how it will be done / what it will encompass. (can also reference to the conclusion or vice versa)

A statement has been added to the beginning of the section to clarify the role of the MMI. Further detail about the role of the MMI has been added to the introduction and conclusion as well.

Priority E could be linked to B re restoration and management. Also the threat to mangrove health isn't clear - the introduction briefly mentioned pollution, but not clear what the threats to mangrove health are and how knowledge of microbiomes are critical to that understanding.

We have added a sentence to the beginning of this section to mention specific threats, which ties it into section P2 (previously B) - lines 164-166.

“Mangrove ecosystems are threatened by a multitude of stressors, such as pollution, sea level rise, coastal development, and sediment salinization. There is an urgent need to better understand the impacts of these stressors on mangroves and their microbiomes.”

Priority F: Following on the above comments, the health connections looks like it's covered more here, so again integration across the priorities would be good to have. Here I see connections to B that can be made re restoration - or perhaps the case for restoration is more fitting in F. Since restoration and

rehabilitation is an ongoing theme, perhaps it can be its own priority if the authors decide to broaden the title and scope of the paper.

This comment is addressed by the restructuring into research priorities and approaches.

L173-175: can a reference be added here or is this unpublished?

The reference has been added:

30. Soldan R, Mapelli F, Crotti E, Schnell S, Daffonchio D, Marasco R, Fusi M, Borin S, Cardinale M. 2019. Bacterial endophytes of mangrove propagules elicit early establishment of the natural host and promote growth of cereal crops under salt stress. *Microbiol Res* 223–225:33–43.

Conclusions: It seems like MMI has an established team which are mostly the co-authors - is this the mentioned collaborative network and closed to people joining, or is there a call for more collaborators or data using the framework outlined in D to build research capacity, questions and datasets?

We have added a sentence in the conclusion to specify that others are welcome to join (lines 235-237):

“We welcome any interested researchers working on the mangrove microbiome to join our network through our website.”

Figure 1: Suggest having the description of E in one section to improve interpretation

We have reorganized the text box to be easier to read.

September 30, 2020

Dr. Sarah M. Allard
University of California San Diego
Department of Pediatrics
9500 Gilman Drive #0202
La Jolla, CA 92093

Re: mSystems00658-20R1 (Introducing the Mangrove Microbiome Initiative: Identifying microbial research priorities and approaches to better understand, protect, and rehabilitate mangrove ecosystems)

Dear Dr. Sarah M. Allard:

Both reviewers have now looked at your revised manuscript. They were both pleased with this new version and I am delighted to accept it for publication in mSystems as I believe that it will be a great contribution to the field.

Your manuscript has been accepted, and I am forwarding it to the ASM Journals Department for publication. For your reference, ASM Journals' address is given below. Before it can be scheduled for publication, your manuscript will be checked by the mSystems senior production editor, Ellie Ghatineh, to make sure that all elements meet the technical requirements for publication. She will contact you if anything needs to be revised before copyediting and production can begin. Otherwise, you will be notified when your proofs are ready to be viewed.

Sincerely,

Jean-Baptiste Raina
Editor, mSystems

Journals Department
Phone: 1-202-942-9338